# The Triangle Wave Versus the Cosine: How Classical Systems Can Optimally Approximate EPR-B Correlations

**DOI:** 10.3390/e22030287

**Published:** 2020-02-29

**Authors:** Richard David Gill

**Affiliations:** Mathematical Institute, Leiden University, Niels Bohrweg 1, 2333 CA Leiden, The Netherlands; gill@math.leidenuniv.nl

**Keywords:** singlet correlations, twisted Malus law, EPR-B experiments, local hidden variables, spinning coloured disk model, spinning coloured ball model, simulation models, Bell’s theorem

## Abstract

The famous singlet correlations of a composite quantum system consisting of two two-level components in the singlet state exhibit notable features of two kinds. One kind are striking *certainty relations*: perfect anti-correlation, and perfect correlation, under certain joint settings. The other kind are a number of *symmetries*, namely invariance under a common rotation of the settings, invariance under exchange of components, and invariance under exchange of both measurement outcomes. One might like to restrict attention to rotations in the plane since those are the ones most commonly investigated experimentally. One can then also further distinguish between the case of discrete rotations (e.g., only settings which are a whole number of degrees are allowed) and continuous rotations. We study the class of classical correlation functions, i.e., generated by classical physical systems, satisfying all these symmetries, in the continuous, planar, case. We call such correlation functions *classical EPR-B correlations*. It turns out that if the certainty relations and rotational symmetry holds at the level of the correlations, then rotational symmetry can be imposed “for free” on the underlying classical physical model by adding an extra randomisation level. The other binary symmetries are obtained “for free”. This leads to a simple *heuristic* description of all possible classical EPR-B correlations in terms of a “spinning bi-coloured disk” model. We deliberately use the word “heuristic” because technical mathematical problems remain wide open concerning the transition from finite or discrete to continuous. The main purpose of this paper is to bring this situation to the attention of the mathematical community. We do show that the widespread idea that “quantum correlations are more extreme than classical physics would allow” is at best highly inaccurate, through giving a concrete example of a classical correlation which satisfies all the symmetries *and* all the certainty relations *and* which exceeds the quantum correlations over a whole range of settings. It is found by a search procedure in which we randomly generate classical physical models and, for each generated model, evaluate its properties in a further Monte-Carlo simulation of the model itself.

## 1. The Problem, in a Picture

Just about every introduction to Bell’s (1964) theorem [1], stating the incompatibility of quantum mechanics with classical physics, contains the picture shown in Figure 1.

The accompanying text claims that the triangle wave (red) is the prediction of local realism, the negative cosine curve (blue) is the prediction of quantum mechanics. The correlations pictured here are known as the singlet correlations, or, after the famous papers Einstein, Podolsky and Rosen (1935) [2] and Bohm and Aharanov (1957) [3], EPR-B correlations. However, the triangle wave is just one of many *possible* correlation functions allowed by local realism. (Similarly, quantum mechanics also allows many more correlations). It is often claimed that quantum correlations are more extreme than classical, but this seems to imply that the “triangle wave” exhibits the most extreme correlations possible in classical physics. Is that true? In what sense might it be true? I am unaware of a clean mathematical answer to this question. The following notes make some attempt to describe concisely all that local realism allows when some key features of the curve are insisted on. We will obtain some answers, some of them new, but also expose many open problems.

The picture shows the correlations obtained when two parties measure something called “spin”, each on one of two correlated (or entangled) particles; they can measure in any direction in certain planes; their measurement outcomes are binary (spin up or spin down in the chosen directions), encoded numerically as ±1. The *x*-axis (horizontal axis) of the graph represents the difference between their setting angles, taken to run from −π to +π radians. The *y*-axis (vertical axis) of the graph, representing the correlation, runs from −1 to +1. Now, the word “correlation” means something different in different fields of science. Physicists often use the word “correlation” to stand for the mean value of the product of two variables. Statisticians might think of the Pearson correlation coefficient, which is obtained after centering the variables concerned by subtracting their mean values and normalising them by dividing by their standard deviations. The physicist’s correlation is the statistician’s raw product-moment. In the present case we consider variables which take the values ±1, and which moreover turn out to take those values with equal probabilities 0.5. It follows that their mean values are zero and their standard deviations equal +1. Statisticians’ and physicists’ correlations coincide.

In the rest of the paper we will see more graphs of correlation functions and in all cases they will be enclosed in the same box. Correlations are plotted against the differences between angles. The box is the rectangle [−π,π]×[−1,+1]. Notice that both curves exhibit perfect correlation when the settings are opposite, perfect anti-correlation when the settings are equal. The sign of the difference between the setting angles is irrelevant. At ±π/2, correlations are zero.

In theory one could also measure in directions in 3D space, and one can ask the corresponding new mathematical question. That question was also formulated, and partially answered, by Kent and Pitalúa-García (2014) [4]. The first version of the present paper was posted to *arXiv.org* in 2013, at the same time as the *arXiv.org* preprint of the Kent and Pitalúa-García paper. There seem to have been almost no further developments since then. As far as we know, the only exception is the much overlooked but very interesting, and it seems to this author, entirely correct, paper by Wang (2014) [5], which apparently never made it past *arXiv.org*. That paper is related to a very interesting earlier paper by Toner and Bacon (2003) [6] which quite satisfactorily solves a somewhat related but very specific problem: how many bits of classical communication are needed for two classical agents, Alice and Bob, to reproduce the 3D quantum correlations? The answer turns out to be just one bit, on average. According to the authors, the required communication protocol is “simple”. Making use of the *detection loophole* to classically simulate quantum correlations is also a case of using, in effect, classical communication, namely by rejecting some *trials*: Alice and Bob are allowed to “try all over again” if either experiences an adverse event in a given *trial* (each has chosen one setting, each has observed one outcome). One wants protocols such that each of the two parties ends up with an outcome *and* each knows that the other indeed has an outcome, too. The various problems—faking *all* the singlet correlations, either with settings in $S^{1}$ or with settings in $S^{2}$, either with discrete or with continuous settings, and doing it with the detection loophole, or with classical communication, or without any cheating at all, but just optimising a sensible expression of the numerical quality of the approximation—are very subtly related. One would like to convert a solution of one problem to a solution of another, but all this is still little understood.

We deliberately do *not* go into the issue of optimality or extremal properties connected to the Bell-CHSH inequality or the Tsirelson inequality (for those, see our other references). Our focus is on the whole curves in Figure 1, not to the *particular* optimisation problems in whose solution they historically arose. Another question yet again, is to discover in what sense the negative cosine is the most extreme that quantum mechanics allows. It is now well known that maximal entanglement is not necessarily a guarantee of maximal non-classical properties.

Recent work by Palmer (2019) [7] suggests that one should take account of the fact that not all settings can be implemented (or cannot be determined—a subtle distinction). Palmer sees a solution in chaos theory, fractals, complexity, and p-adic analysis. We feel that this particular direction of research is a red-herring, though mathematically intriguing. In Bell experiments, we would argue, an apparatus essentially allows an experimenter a discrete collection of buttons to press. What is going on inside the measurement device is irrelevant. Whether the buttons (or levers, or dials) implement exactly or approximately (and in what sense) true directions is irrelevant to Bell’s theorem. As far as the experimenters are concerned, we may pretend that experimenters only have a choice of, for example, 360 different settings. Maybe, nowadays, 360×60×60. So what? We also know for sure that correlations are never perfect. But it is certainly mathematically interesting to study what is mathematically possible with continuously varying settings, and what is mathematically ruled out if perfect correlations were a reality. A second step is to investigate how sensitive are solutions, in terms of sensible measures of sensitivity, to failure of the idealised assumptions. Bell (1965) [1] already devoted a little read section at the end of his famous paper to that question. *All* models are wrong but some can still be useful.

## 2. The Problem, Formalised

According to quantum mechanics, it is in principle possible to arrange the following experiment. Alice and Bob are in their respective laboratories at distant locations, but have set up all kinds of practical arrangements in advance. In particular, they are at rest with respect to the same inertial frame of reference and they have set up accordingly synchronised clocks. They both possess some kind of random number generators and are able to simultaneously and independently choose angles α and β in the interval [0,2π) according to any desired probability distributions. They each input their chosen angle into a physical device in each of their laboratories and after a short time interval, the device responds with a binary output, which we shall code numerically as ±1. The length of time between initiating the choice of random angle and output of ±1 is so short that a signal travelling at the speed of light and carrying Alice’s chosen angle from Alice’s to Bob’s lab could not arrive till after Bob’s output is fixed, and vice-versa. We call the inputs *settings* and the outputs *outcomes*.

This can now be repeated independently as many times as one likes, say *N* times, resulting in synchronised lists, all of length *N*, of settings (angles) and outcomes (±1) at the two locations. We call these *N* repetitions *runs*.

In an ideal experiment, the outcomes of each separate run (a pair of random numbers ±1) are statistically independent from those of other runs and distributed according to the following conditional probability law (conditional on the chosen settings α and β):$$\mathrm{Pr}\left(++\right)\;=\;\mathrm{Pr}\left(--\right)\;=\;\frac{1}{4}\left(1-\cos(\alpha -\beta )\right),$$
$$\mathrm{Pr}\left(+-\right)\;=\;\mathrm{Pr}\left(-+\right)\;=\;\frac{1}{4}\left(1+\cos(\alpha -\beta )\right).$$
These joint probabilities are a manifestation of the *twisted Malus law*.

Notice that these joint probabilities possess a large number of symmetries: they are symmetric with respect to rotation, parity switch (exchange of outcome values ±1), exchange of the two parties (Alice and Bob), and chirality switch (exchange of clockwise and anti-clockwise). They also reflect two “certainty” relations: at exactly equal settings, outcomes are opposite with probability one; at exactly opposed settings, outcomes are equal with probability one. As a consequence of the symmetry in outcome values, we also have a “complete randomness” property: each outcome separately, whatever the setting, is a symmetric Bernoulli trial with outcomes {−1,+1}.

## 3. Classical Physical Representation

If these outcomes were generated by a classical (i.e., local realist) physical model, we would be able to construct simultaneously defined random variables A(α) and B(β), α, β∈[0,2π), such that in one run of the experiment, all these random variables are realised simultaneously, and the actually observed outcomes are merely selected by the *independent* choice of settings α, β∈[0,2π). It seems physically reasonable to assume that the joint probability distribution of the complete stochastic processes *A*, *B* satisfies the same symmetries: i.e., the symmetries observed in the twisted Malus law reflect underlying (physical, fundamental) model symmetries.

This raises measurability issues. We have introduced an uncountable collection of random variables. What do we mean by “the complete stochastic processes *A*, *B*”? It means that we need to make assumptions of their *sample paths*. We need to discuss properties of the indexed set of all (A(α,ω),B(β,ω)), thought of, together, as a *random function*: its values are a function of $\left(\alpha ,\beta \right)\in S^{1}\times S^{1}$ with values in ${\left\{-1,+1\right\}}^{2}$, which is random through its dependence on an underlying ω∈Ω. However, this paper is not going to delve into measurability issues, to which there are several approaches in the mathematical literature on stochastic processes; we will only make a few heuristic remarks on that topic. What is a fact, is that under obvious measurability assumptions, we can symmetrise a given model, converting a non-symmetric model to a fully symmetric one: this is because probabilistic mixing of stochastic processes with the same, given, marginal distributions, results in a new stochastic process with the same marginal distributions. We will explicitly impose the symmetry under rotations. As we will see, because of the “certainty relations” which we also impose, the other symmetries in the correlation functions are automatically true. So whether or not the other symmetries are imposed on the underlying process makes no difference to the family of correlation functions which can be generated by the model. In particular, if we restrict measurement settings (by which we mean the external inputs, not the physical direction which we imagine actually realised in a mathematical model of a physical apparatus) to one of just 360 possible whole-degree settings on a digital dial, no measurability assumptions are needed whatsoever.

However, even if we like to think of external inputs as continuous, we would like here to offer a heuristic argument for assuming some “niceness” of the sample paths of (A,B). We have already assumed, as part of the meaning of local realism, the mathematical existence of a single probability space on which are defined two indexed families of binary random variables A(α) and B(β). It follows that the joint probability distributions of any finite number of these random variables are also well defined. The random function A(α),α∈[0,2π) takes values in {−1,+1} but might in principle be extraordinarily irregular. We propose *assuming* that it can be taken to be a piece-wise constant function making only a finite number of changes of value: in other words, {α∈[0,2π):A(α)=1} is a finite union of intervals, and so of course is its complement too, {α∈[0,2π):A(α)=−1}; and the same for *B*. That finite number can be random, and it can be arbitrarily large! Here is some support for the assumption. To begin with we would argue that there is no loss in *physical* generality in assuming that any realisation of the set {α∈[0,2π):A(α)=1} is a Borel measurable subset of [0,2π). A mathematical argument for this physical claim could run as follows: if one is prepared to reject the axiom of choice, it is possible to axiomatically demand that all subsets of the real line are measurable. (The mathematical existence of non-measurable sets is entirely a matter of mathematical taste, it refers to how large or small we want our abstract mathematical universe to be. Even with the axiom of choice, though they exist mathematically, non-measurable sets cannot be constructed or computed or exhibited in any sense.) We may therefore just as well assume the realisations of the just mentioned sets are at nice enough that they are Borel measurable. Now, an elementary result in measure theory is that any bounded measurable subset of real numbers can be arbitrarily well approximated by finite unions of intervals in the following sense: for any ϵ>0 one can find an approximating set (a finite union of intervals) such that the set-theoretic difference between the set being approximated and its approximation can be covered by a countable set of intervals the sum of whose lengths is at most ϵ (the axiom of choice is not needed to establish this result). Thus, there is no loss of generality is assuming that set {α∈[0,2π):A(α)=1} can be well enough approximated by a finite union of intervals, and similarly for *B*. Notice, that the number of intervals which give a good enough approximation can be arbitrarily large.

We will therefore simply proceed (in the case of continuous settings) after making the following *assumption*: the stochastic processes *A* and *B* will be assumed to be (to a good enough approximation) such that the sets of angles where they take their possible values ±1 are finite unions of intervals. It is easily shown from this that the angles at which the value jumps between ±1, and the number of those angles, are random variables (they can be written as limits of functions of each of only finitely many coordinates). Now think of the sample paths of *A* and *B* as two functions on the unit circle. Because of one of the “certainty relations” between *A* and *B* it follows that the sample path of *B* is identical to the path of *A* after rotation through an angle π. It follows from the other that the sample path of *B* is identical to the negative of the path of *A*. Thus, the two sample paths are determined completely by the path of *A* on the first half of the circle, [0,π). The negative of the same path is repeated, for *A*, on [π,2π); the path of *B* is the path of *A* shifted (rotated) a distance π.

Let us furthermore suppose that the joint probability distribution of *A* and *B* is invariant under rotation, just as the correlation function is. This assumption can be made “without loss of generality”, because if the original correlation function of the original classical model has this invariance, then a new classical model with the same invariance can be built simply by choosing, uniformly at random, a “rotation” of the original model. See also [4]. Now, we can write the joint probability distribution of the processes *A* and *B* as a probabilistic mixture over the (finite) numbers of jumps of each process. Any such probability distribution can be built up as follows, according to what I call the spinning randomised bi-coloured disk model. First we look at the spinning deterministic bi-coloured disk.

## 4. The Spinning Bi-Coloured Disk

Take a fixed even number k≥0 and fixed angles $0<\theta_{1}<\cdots <\theta_{k}<\pi$. Colour the k+1 segments $\left(0,\theta_{1}\right)$, $\left(\theta_{1},\theta_{2}\right)$, …, $\left(\theta_{k},\pi \right)$ “black”, “white”, …, “black”. If k=0 there is just one segment (0,π), and it is coloured black. Colour (π,2π) in complementary way: $\left(\pi ,\pi +\theta_{1}\right)$, …, $\left(\pi +\theta_{k},2\pi \right)$ are coloured “white”, “black”, …, “white”. The colour assigned to end-points does not matter, choose some suitable convention. We have now coloured the entire unit circle with our two colours “white” and “black’, alternating finitely many times, such that each point is opposite a point of the opposite colour. The total length of white segments and the total length of black segments are therefore equal. Now give the coloured unit circle a random rotation chosen uniformly between 0 and 2π. Define A(α)=±1 according to whether the colour of the randomly rotated circle at point α is black or white. Define *B* as the rotation of *A* through the angle π. The marginal distributions of *A* and *B* are fair Bernouilli trials, outcomes ±1, also known as Rademacher random variables. Their joint distribution is invariant under exchange of *A* and *B*, and under simultaneous switch of both their signs, and automatically also under sign change of the difference between their angles.

## 5. The Randomised Spinning Bi-Coloured Disk

If we choose *k* at random according to an arbitrary probability distribution over the even non-negative integers, and then $0<\theta_{1}<\cdots \theta_{k}<\pi$ according to an arbitrary joint distribution given *k*, colour endpoints of the intervals according to a consistent convention, and finally choose a rotation of the coloured circle completely at random, we have defined two stochastic processes *A* and *B* (possessing all desirable measurability properties), such that the physical model for *A* and *B* is invariant under rotation, possesses the desired “certainty relations”, and exhibits sample paths which have are piecewise constant with only finitely many jumps. This recipe generates the class of all processes *A* and *B* subject to rotation invariance, certainty relations, and regularity of sample paths (finitely many sign changes). It therefore generates all of these process’s correlation functions.

An open, partly mathematical and partly physical or meta-physical problem, is to investigate the possibility to extend this class of processes in an appealing way by some limit procedure or closure operation, so as to add processes with less regular paths, but without extending the class of correlation functions which they generate. To solve this problem requires both mathematical thought, expertise in the foundations of mathematics, and physical or meta-physical (philosophical) argument. We hope others will rise to this challenge.

Marginally, each A(α) and B(β) represents a symmetric Bernoulli trial—each is a Rademacher random variable. The joint distribution of A(α),B(β) for any fixed pair of angles (α,β) possess the rotational invariance, the certainty relations, and the three binary symmetries. This joint probability distribution consists of four probabilities adding to one. Since its two margins are symmetric Bernoulli trials, the joint distribution is completely determined by the single number Pr(A(α)=B(β))=Pr(A(α)≠A(β) which is the probability that the number of colour switches on the randomly rotated coloured unit circle between angles α and β is odd. This probability can only depend on the *absolute value* of the difference between these angles, hence is invariant under *parity* switch (switching the sign of both outcomes), *party* switch (exchanging Alice for Bob) and *chirality* switch (measuring angles clockwise or anti-clockwise).

The raw product moment between A(α) and B(β) is the expectation of its product, and is easily seen to be equal to 2Pr(A(α)=B(β))−1. Within our local realist model, B(β)=−A(β) so that is the same as 2Pr(A(α)≠A(β))−1. If we write δ=β−α then we obtain from this the correlation function ρ(γ)=2Pr(A(γ)≠A(0))−1=2Pr(A(−γ)≠A(0))−1.

## 6. Computation

The arguments given so far show that, if we restrict ourselves to a class of sufficiently regular classical physical models, the set of all possible classical correlation functions ρ is the set of all convex combinations of correlation functions corresponding to a spinning bicoloured disk model determined by an even number k≥0 and some $0<\theta_{1}<\cdots \theta_{k}<\pi$. To each *k* and $\theta_{1}$, …, $\theta_{k}$ there corresponds a colouring of the unit circle, described above. This colouring determines stochastic processes *A* and *B*. Their correlation function ρ can be described in terms of the colouring of the unit circle as follows: pick a point uniformly at random on the unit circle. Then ρ(γ) is the probability that the uniform random point has the opposite colour (black or white) to that of the point at a distance γ clockwise around the circle from the first chosen point.

It is an open problem to further investigate whether the class of “sufficiently regular classical physical models” can be meaningfully extended from classical models such that the sample paths of *A* and *B* only make finitely many sign changes to classical models such that the sample paths of *A* and *B* can only be approximated by such nice functions.

In Figure 2 we see a sample of 12 such correlation functions, all with k=4, but with various values of $\theta_{1}$ to $\theta_{4}$. That means that the basic spinning disk always has 10 segments. Each plot includes a picture of the spinning disk which generate the curve. Completely coincidentally, the first graph shows a spinning disk model which almost reproduces the triangle wave, even though it has ten segments, rather than two. The second graph on the other hand, also by coincidence, shows some correlations which are larger than the cosine correlations where both are positive. This also happens in two more of the twelve plots. That is rather unusual, it can also easily happen that in twelve random plots there are no “violations”.

The plots were made using the programming language R, the code is contained in the Appendix A. The reader is invited to replace the assignment nswitch <−4 (“nswitch gets the value 4”) by assignments of other even numbers and look at the results. The value 0 leads straight to the triangle wave. The value 2 seems to lead to curves which nowhere exceed the triangle wave. At nswitch equal to 4, roughly one in ten random disks has an interval of positive length of angle differences where the correlation exceeds the negative cosine (where both are positive). At larger values still, the whole curve shows more oscillations, and gets overall closer and closer to flat (correlation 0) except at the locations where it is always forced to equal ±1, where its peak (or downwards spike) necessarily gets sharper and sharper.

Notice that all of these “curves” are actually piecewise linear. Given a particular disk colouring, it is not a great problem to figure out the knots and slopes which together determine the curve, but the author of this paper trusted more in his programming skills to get the right answer quickly and accurately enough by computer simulation.

The second plot of the display of 12 shows the most clear “violation” of the folk-lore result that the triangle wave is the best that classical physics can do. Indeed, we see there a spinning disk model such that some of the positive correlations are clearly much larger than the single correlations. We reproduce it in Figure 3.

Note that a variation of the bi-coloured spinning disk model is a tri-coloured spinning disk, where an intermediate colour “grey” actually means “invisible”: the measurement of spin fails, and instead the observer gets a report “there was no detection”. In many past real experiments, non-detections were a massive problem. If trials are simply discarded where either particle was not observed at all, it turns out that one can actually perfectly reproduce the negative cosine if there are enough “no-detections”, even though the model is completely classical. Another alternative is to keep the “no-detections”, but to score the outcome in those cases as 0. This reduces the amplitude of the correlation curve. Again, it turns out that one can perfectly reproduce a negative cosine but with a lower amplitude, if there are enough “no-detections”. A negative cosine does not in itself prove a quantum origin of the observed correlations!

## 7. Conclusions: What Next?

This note is more of a research proposal than a report of definitive results. There are two main directions to explore, mathematically. One direction concerns the investigation of regularity conditions and approximation. I think it is mathematically important to tidy up the details but I do not think this direction is physically interesting. It might be foundationally interesting, it could certainly be mathematically challenging. But I do not find it physically interesting, because of the following argument. Suppose we only considered *intended* measurement angles, settings which can be chosen by external agents, and the menu of settings which can be chosen are whole numbers of, say, degrees, minutes and seconds. What angles actually result inside the black-box, when a particular setting is chosen on the outside, is irrelevant. It is part of the physical system of which we are interested in whether or not it has a classical physical description. Then there are no regularity issues at all. The representation of all classical physical EPR-B correlations as the correlations of a a random spinning disk model is mathematically precise. The lengths of the segments of the disk of different colours are restricted to be whole numbers of seconds. We only investigate the correlation functions at whole numbers of seconds. Thus, we finish up looking at a slightly smaller class of correlation functions, and we only look at them on the very fine lattice of “whole second” angles. For practical purposes, we will see no difference with the results of the “continuous” analysis given in this paper.

I think a second direction is more interesting. Let us accept the class of correlation functions arising from the spinning disk model. Can we analytically describe this class of functions, or the topological closure of this class of functions (according to a convenient but meaningful topology) in an alternative succinct way? For instance, is there an elegant description of the characteristic functions of these correlation functions?

An intriguing possible direction involving indeed the Fourier transform is suggested by some lecture slides by Steve Gull, going back now 35 years. On his “MaxEnt 2009” web-page http://www.mrao.cam.ac.uk/~steve/maxent2009/, under the heading *Quantum Acausality and Bell’s Theorem*, Steve writes
Many years ago (about 1984), I used to give a Mathematical Physics course to the Part II students. I illustrated the quantum paradox covered by Bell’s theorem by showing that you can’t program two independently running computers to mimic the results of spin measurements on two spin-1/2 particles in a singlet state. I believe this demonstration is actually better than Bell’s original argument.
His slides can be found at http://www.mrao.cam.ac.uk/~steve/maxent2009/images/bell.pdf. They sketch a lovely proof of Bell’s theorem using the fact that the Fourier transform of the correlation function ρ has to equal the expected squared absolute value of the Fourier transform of the random function *A*. The actual correlation function of the negative cosine only has three non-zero Fourier coefficients. However, the Fourier transform of any realisation of *A* must have infinitely many non-zero coefficients, since otherwise it could not have any jumps. Since their absolute values get squared before averaging, there is no way that all but three can vanish.

I have two ideas where to go next. Firstly, if we insist that the correlation function not only has the symmetries we want, but is also monotone decreasing (between 0 and π), that will further narrow the possibilities. Maybe there is only one left—the triangle wave of my first picture. Classical correlations can exceed quantum correlations but, it seems, only at the cost of oscillations. Thus, the idea is to add one *qualitative* condition: monotonicity. Secondly, we can express the $L_{2}$ distance between two curves in terms of the $L_{2}$ distance between their Fourier transforms: might that give us a way to show that the triangle wave is the closest approximation to the cosine?

## Figures and Tables

**Figure 1 entropy-22-00287-f001:**
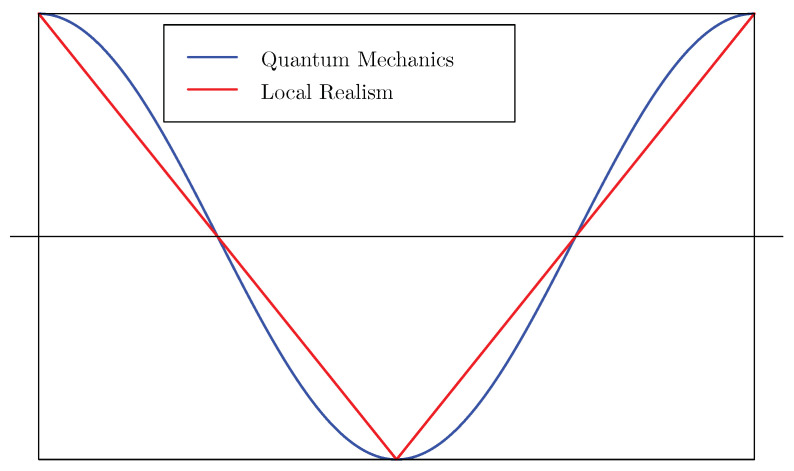
Correlation, between −1 and +1, plotted against angle between measurement directions, from −π to π.

**Figure 2 entropy-22-00287-f002:**
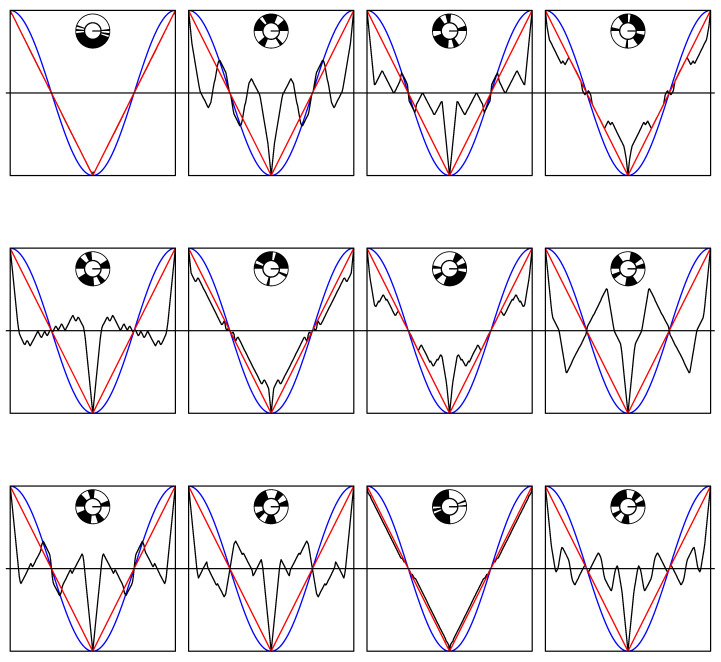
12 EPR-B correlations (in black), each determined by a (randomly generated) spinning coloured disk of 10 segments.

**Figure 3 entropy-22-00287-f003:**
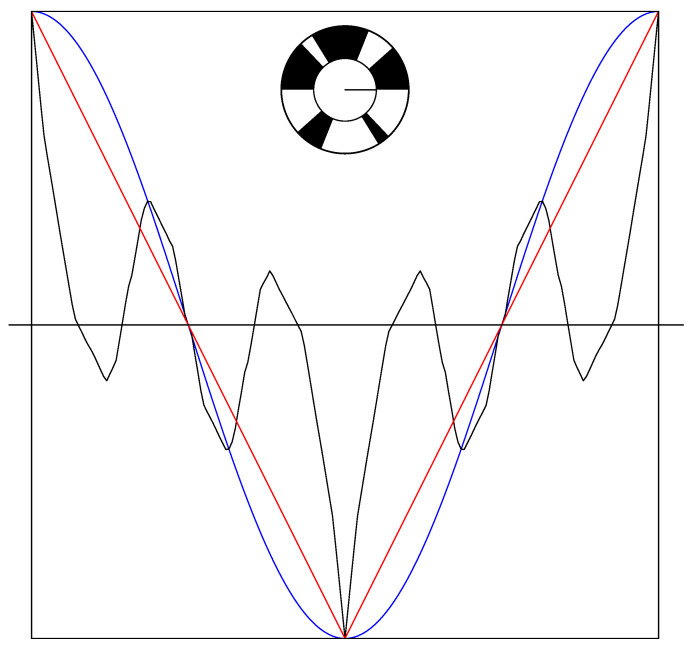
The second spinning disk model of the 12 in Figure 2.

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
