# Peer review of "The Triangle Wave Versus the Cosine: How Classical Systems Can Optimally Approximate EPR-B Correlations"

_entropy, 2020, doi:10.3390/e22030287_

Round 1

Reviewer 1 Report

As the author himself points out, this manuscript is more like a research note or a precursor to a research problem than a technically written scientific paper. I actually find it somewhat difficult to figure out exactly what the author is trying to say here: there are multiple unclear or confusing sentences and statements in the paper concerning the mathematical definitions (such as the "correlation function") model(s) and the meaning of the results.

I cannot recommend publishing this note as is. It could be an interesting paper but only after rewriting and explaining all the different quantities, models, equations, and the meaning of the results in more detail.

Author Response

The reviewer wrote "It could be an interesting paper but only after rewriting and explaining all the different quantities, models, equations, and the meaning of the results in more detail." I agree, the paper was definitely "unfinished" and, moreover very confusingly written. I have done, I believe, exactly what you wanted: I did a lot of rewriting and explaining. I also discovered some new results and also found some further, relevant references. 

The arguments in the paper are subtle. There are physical issues, and meta-physical issues, as well as purely mathematical issues. I am a mathematician, and neither a philosopher nor a physicist. I'm also no expert in some relevant parts of mathematics (deep foundational issues). Despite this, I think that I can usefully point to several serious and interesting and important mathematical open problems, which have been neglected for too long. The purpose of the paper is to bring them to the attention of the relevant scientific communities. There is also a pervasive but false myth to be dispelled, namely that quantum correlations are somehow stronger than classical correlations. Probably, the real experts know that this is not really true, but there is a lot of published "mis-information" about Bell's theorem, even propagated by authoritative voices. I hope that my paper can offer some clarification.

Reviewer 2 Report

See please the attached file with recommendations. 

Author Response

Thank you for your appreciation of my paper. You wrote "Moderate English changes required" and I certainly do agree that that was the case. The other referee found my paper pretty incomprehensible but he or she did have the feeling that it might be useful if only it could be rewritten with more clarity.

I have now carried out a major rewriting of the paper, and moreover, redid the computations and the programming, discovered a new result which I am excited about, and also discovered very relevant literature references. 

Round 2

Reviewer 1 Report

The author has greatly improved the paper by adding more details and explaining things more clearly. It would still help the reader if he carefully defined the "correlation function" shown in multiple figures in the manuscript. This term has different meaning in different fields and should be carefully defined. Related to this the author should follow standard scientific practice and add labels and numbers (ticks) the all the figures showing correlations.

Author Response

The reviewer writes "It would still help the reader if he carefully defined the "correlation function" shown in multiple figures in the manuscript. This term has a different meaning in different fields and should be carefully defined. Related to this the author should follow standard scientific practice and add labels and numbers (ticks) the all the figures showing correlations."

I have rewritten Section 1 of the paper to explain exactly what I mean by "correlation". In the same section, I have also given a more extensive description of all the figures in the paper. I experimented with adding labels, numbers, and ticks to all the figures. This cost me a day of work at the computer, but in my opinion it did not improve the paper at all. All graphs, deliberately, have the same axes; they represent the same variables varying between the same limits, represented as a rectangular box. The text now makes absolutely clear what those limits are. A vertical axis going through the middle of the graph together with ticks, numbers, and labels on both axes, would hide from the readers' view what I want them to see in the graphs.

Standard scientific practice is standard practice for very good reasons. But in this case, I think the usual reasons do not apply.